# Protein Adsorption onto Modified Porous Silica by Single and Binary Human Serum Protein Solutions

**DOI:** 10.3390/ijms22179164

**Published:** 2021-08-25

**Authors:** Diego R. Gondim, Juan A. Cecilia, Thaina N. B. Rodrigues, Enrique Vilarrasa-García, Enrique Rodríguez-Castellón, Diana C. S. Azevedo, Ivanildo J. Silva

**Affiliations:** 1Centro de Tecnologia, Grupo de Pesquisa em Separações por Adsorção—GPSA—Departamento de Engenharia Química, Campus do Pici, Bl. 709, Universidade Federal do Ceará, Fortaleza 60455-760, CE, Brazil; diegoromao19@yahoo.com.br (D.R.G.); thaina.nobre@hotmail.com (T.N.B.R.); enrique@gpsa.ufc.br (E.V.-G.); diana@gpsa.ufc.br (D.C.S.A.); ivanildo@gpsa.ufc.br (I.J.S.J.); 2Departamento de Química Inorgánica, Cristalografía y Mineralogía, Facultad de Ciencias, Campus de Teatinos, Universidad de Málaga, 29071 Malaga, Spain; castellon@uma.es

**Keywords:** purification, adsorption isotherm, human serum: porous silica, SBA-15, mesocellular foam

## Abstract

Typical porous silica (SBA-15) has been modified with pore expander agent (1,3,5-trimethylbenzene) and fluoride-species to diminish the length of the channels to obtain materials with different textural properties, varying the Si/Zr molar ratio between 20 and 5. These porous materials were characterized by X-ray Diffraction (XRD), N_2_ adsorption/desorption isotherms at −196 °C and X-ray Photoelectron Spectroscopy (XPS), obtaining adsorbent with a surface area between 420–337 m^2^ g^−1^ and an average pore diameter with a maximum between 20–25 nm. These materials were studied in the adsorption of human blood serum proteins (human serum albumin—HSA and immunoglobulin G—IgG). Generally, the incorporation of small proportions was favorable for proteins adsorption. The adsorption data revealed that the maximum adsorption capacity was reached close to the pI. The batch purification experiments in binary human serum solutions showed that Si sample has considerable adsorption for IgG while HSA adsorption is relatively low, so it is possible its separation.

## 1. Introduction

The selective immobilization and releasing of biomolecules, such as proteins, enzymes or oligonucleotides in porous structures and/or on the surface of nanoparticles, has emerged as an interesting process in several fields, such as biocatalysts [1,2,3], bioengineering [1,2,3], biosensing [4,5], separation by chromatographic techniques [6,7,8] or drug delivery [9,10,11]. Many adsorbents have been used for encapsulation of proteins, such as layered double hydroxides (LDH) [12,13,14,15,16], hydroxyapatites (HA) [17,18,19], polysaccharides and nanoparticles-encapsulated in polysaccharides [20,21,22,23,24], chitosan [25,26,27,28,29], some polymers and polymeric nanoparticles [30,31,32,33,34,35,36,37,38,39], and mesoporous silicas [7,40,41,42,43,44,45,46].

Focusing on ordered mesoporous silicas, the M41S (MCM-41, MCM-48, MCM-50) family [47] and the SBA (SBA-15, SBA-16) family [48,49] have been highly studied in adsorption and catalysis processes since its pore diameter can be modulated. Between them, the SBA-15 displays greater wall thickness than the MCM-41, which supposes a mesoporous structure with higher thermochemical and mechanical stability than the MCM-41 [48,50,51,52].

In the case of the SBA-15, the diameter of the mesochannels can be also modified by adding pore expander agents, such as alkanes [53,54], amines [55] or aromatic compounds [53,56]. This fact minimizes the diffusional problems of the SBA-15 mesochannels. In addition, these problems can be also minimized by the addition of fluoride species in the synthetic step since fluoride limits the polymerization of the silica, obtaining porous structures with shorter channels denoted as mesocellular foam (MCF) [53,57] leading to versatile materials.

The modulation of the pore diameter allows to host or encapsulate from small metal particles to proteins or nucleic acids. The encapsulation capacity of biomolecules on porous silica has been evaluated in the literature using several target molecules, such as lysozyme [41,44,45,46], cytochrome c [58,59], myoglobin [60], hemoglobin [61] or bovine serum albumin (BSA) [42,45,46], whose dimensions and composition varies significantly. It is necessary to know the dimensions of each biomolecule to design a porous structure with adjustable pore diameter and appropriate pore length to favor the adsorption process and its subsequent elution [40,43]. The interaction between the biomolecules and the porous silica must be weak since a strong interaction between the biomolecule and the porous silica implies more severe elution conditions, which could imply the denaturation of the biomolecule [62]. Thus, the most appropriate interaction to recover biomolecules is the physical adsorption composed by the sum of non-covalent interactions as van der Waals or hydrogen bonds together with electrostatic interactions [63]. In this sense, the ionic strength and the pH must be adjusted to minimize the repulsive effects. Thus, the maximum adsorption capacity often takes place when the total charge of the biomolecule is zero. This point is denoted as isoelectric points (pI) [64]. However, biomolecules are very complex structures and, despite having a zero charge, biomolecules can have charged residues where they can have both attractive and repulsive interactions [65].

The adsorption capacity of the porous silica can also be modified by the incorporation of heteroatoms, such as Al [66,67] or Zr [46,68], since the substitution of Si atoms by Al or Zr atoms modifies the electronic density of the porous silica and generates acid sites. This fact can improve the interaction between the porous structure and the biomolecules as previously indicated several authors [66,68].

In the last decades, protein adsorption in mesoporous materials has been extensively studied, and many technical applications have been investigated. Some proteins, when immobilized in mesoporous silicas, present a better chemical and thermal stability, besides retaining or increasing their electrochemical activity even when subjected to denaturation conditions [69]. The advantages of working with these materials allow a wide biochemical application, for example, the use of these porous materials has shown to be very promising as new devices for the controlled release of the drug “in vivo” [70]. Generally, protein adsorption/elution seems to be non-specific and may be reversible under certain conditions. Interestingly, despite the vast amount of research in this field, there is no consensus in the literature on the type of interaction that dominates the adsorption of biomolecules on inorganic surfaces due to the complexity of biomolecules [70].

In spite of the porous silica having been evaluated in a plethora of biomolecules, these adsorbents have not been studied to adsorption of typical human serum proteins, such as human serum albumin and immunoglobulins, yet.

Human serum albumin (HSA) is the most abundant protein in human blood plasma, soluble in water and monomeric. This protein is responsible for the regulation of osmotic pressure, transport of hydrophobic reagents and inactive toxic metabolite [71,72,73]. HSA is synthesized in the liver as preproalbumin, which has an N-terminal peptide that is removed before the nascent protein is released from the rough endoplasmic reticulum [74,75]. The reference range for albumin concentrations in serum is approximately at around 35–52 g L^−1^ [75]. This protein consists of a single polypeptide chain, which contains 585 amino acid residues containing seventeen pairs of disulfide bonds and one free cysteine, and its molecular mass is 66.4 kDa. The dimension of HSA is 9.5 mm × 5.0 mm × 5.0 nm kDa (Appendix A) [71,73,76]. This protein consists of three similar structures divided into two domains. These six mounted helical subdomains series form a molecule in the form of a “heart” [73].

Immunoglobulin G (IgG) is a type of antibody, representing approximately 75% of serum antibodies in humans, being the most common type of antibody found in blood circulation. These antibodies are created and released by plasma B cells. By binding many kinds of pathogens such as viruses, bacteria, and fungi, IgG protects the body from infection [77,78]. Regarding the class of immunoglobulins (Igs), all of them have in common regardless of their specificity, four polypeptide chains: two heavy (H) chains and two light (L) chains [79,80]. The light and heavy chains are joined by a disulfide bond. These disulfide bonds are located on a flexible region of the heavy chain, known as the hinge, a region of approximately 12 amino acids that is exposed to enzymatic or chemical cleavage. All four polypeptide chains contain variable regions (V) and constant (C) found at the amino and carboxy-terminal portions, respectively. The effector functions of antibodies, such as placental transport or antigen-dependent cellular toxicity, are mediated by structural determinants within the immunoglobulin Fc region [81,82,83]. The Igs are classified in five types: IgA, IgD, IgE, IgG and IgM. Each class of Igs has their own specific characteristics, having very similar structures differentiated by its heavy chain. Among the Ig stand out IgGs for representing about 70 to 75% of total serum Ig and play a fundamental role in neutralizing toxins and protecting the fetus, this being the only antibody that crosses the placenta. IgG is formed by a group of glycoproteins and consists of four subclasses (IgG_1_, IgG_2_, IgG_3_ and IgG_4_) [81], variably pI from 6.3 to 9.0 and molecular mass of 150 kDa, with approximate dimensions of 14.5 cm × 8.5 nm × 4.0 nm (Appendix A) [84].

This research proposes the synthesis of mesoporous silicas, which were modified using a pore expander agent and fluoride species to obtain porous silica with high pore diameter and short length of the channels. On the other hand, the influence of the Zr-content incorporated into the porous silica was also evaluated in the adsorption of HSA and IgG. From the adsorption studies, it was identified the pH at which occurs the highest adsorption capacity of each protein. In the next step, the kinetic studies were carried out using phosphate and acetate buffers solutions. Finally, binary mixture with IgG and HSA were mixed in different proportions to evaluate the adsorption capacity by electrophoreses.

## 2. Characterization of the Materials

Figure 1 shows the low-angle X-ray diffraction patterns between 2θ: 0.5–10° for the synthesized materials. Despite the addition of pore expander agents and fluoride species to limit the growing of the porous silica, Si sample still maintains a broad diffraction peak located at 2θ of 1.55°. This peak is ascribed to the d_100_ reflection of the porous silica, which is associated to a hexagonal symmetry (p6mm) [43,46,48]. Unfortunately, other secondary reflections of the SBA-15, as d_110_ and d_200_, cannot be observed. This fact supposes that structure modifying agents influence the long-range ordering of the porous silica, as was reported in the literature [54,56,57]. In this sense, several authors have reported that the interaction between TMB and triblock copolymer favors the expansion of the pore size of the SBA-15 and hence the increase in the volume of the hydrophobic template resulting in a porous structure with a lower degree of long-range ordering. Nonetheless, it is necessary the expansion of the pore diameter in the channels to host bulky molecules such as IgG and HSA in its framework [42,45].

The incorporation of Zr-species in the synthesis step does not worsen the long-range ordering of the porous material since the d_100_ reflection maintains both its position (2θ = 1.55°) and intensity. This fact indicates that despite the alkoxides of Zr and Si have a different hydrolysis rate, the hexagonal symmetry (p6mm) structure remains.

The analysis of the high angle diffractograms (not shown) only displays a broad peak about 2θ = 22°, which is typical of the porous silica walls. On the other hand, diffusion peaks attributed to ZrO_2_ do not appear either, even for the SiZr5 sample, so the segregation of SiO_2_ and ZrO_2_ phases must be ruled out. These data suggest that the Zr-species must be well inserted in the silica matrix or at least well-dispersed on the porous silica matrix [85,86].

To determinate morphology of the porous silica modified with pore expander agents, transmission electron microscopy was carried out (Figure 2). Figure 2A shows that the traditional SBA-15 displays a honeycomb structure. The modification of this structure with fluoride species and TMB as a pore expander agent (Figure 2B) causes a strong modification of the structure, obtaining a less-ordered porous structure, with shorter channels and higher pore diameter. The incorporation of the Zr-species in the synthetic step (Figure 2C–E) seems to increase the disorder of the porous structure leading to poorer defined porous structures.

In order to determine the textural properties of the porous silica and the Zr-doped porous silica, N_2_ adsorption–desorption analyses at −196 °C were carried out (Appendix A). According to the IUPAC classification, all isotherms can be considered as type IV, which are typical of mesoporous materials [87]. From the profile of the N_2_ adsorption–desorption isotherms, it can be observed that the hysteresis loops take place between a relative pressure of 0.80 and 0.95. These values are well above those shown by the traditional SBA-15, where the hysteresis loop appears at a relative pressure of around 0.7 [45,88]. This increase suggests the formation of a porous structure whose pore size is much larger than that shown by the traditional SBA-15.

The analysis of the hysteresis loops of the porous materials shows how the profile of the N_2_ adsorption–desorption isotherm of the porous silica resembles the H3 type, which can be considered as typical of macroporous structures [87]. The incorporation of the Zr-species in the synthetic step causes a slight modification of the hysteresis loops, being classified as an H1 type, which is typical of porous materials with uniform mesopores or/and macropores. In all cases, the hysteresis loops are relatively narrow, which indicate a homogeneous pore size distribution.

The specific surface area was determined from the S_BET_ equation [89] (Table 1). All porous materials show a specific surface area between 330–440 m^2^ g^−1^ and a large total pore volume (1.51–2.54 cm^3^ g^−1^). Despite these values being relatively high, the typical SBA-15 reaches the highest S_BET_ value. In this sense, previous research has reported that the use of pore expander agents causes a decrease in the interaction of the P-123 molecules used as templates in such a way the interconnection between adjacent micelles and subsequent the microporosity and the S_BET_ values also diminish [90]. The incorporation of the Zr-species causes a decrease in the pore volume, which is directly related to the descent of the pore diameter due to the formation of less-ordered porous materials because of the different rate of hydrolysis observed between silicon and zirconium alkoxides [46,85].

The pore size distribution was determined from the NLDFT method (Appendix A) [91]. It can be observed how the pore expander agents favors an increase in the pore diameter than that shown for SBA-15 in the literature, which is around 7 nm [45,88]. In all cases, it is noteworthy the presence of microporous with a maximum of 1.4 nm. Several authors have pointed out that the mesochannels of the porous silica (SBA-15) are interconnected between them [90]. The addition of pore expanding agents as TMB diminishes this connection between adjacent channels although these micropores are still present in these expanded pore materials. In addition, this pore expander agent also causes a clear increase in the average pore size, reaching a pore width whose maximum is located at 25 nm for the Si sample (Appendix A). The incorporation of the Zr-species in the synthetic step leads to a slight decrease at the maximum of the pore distribution since this value diminishes to 20 nm. Considering the pore diameter of the Zr-doped porous silica, it is expected that HSA and IgG can access, at least partially, inside the pore of the adsorbents of the present research since the dimensions of HSA is about 7.5 nm × 6.5 nm × 4.0 nm [76], while the IgG composed by four subclasses of antibodies (IgG_1_, IgG_2_, IgG_3_ and IgG_4_) has dimensions of 14.5 nm × 8.5 nm × 4.0 nm [82].

It has been reported that the incorporation of the Zr-species causes a modification of the electronic density, generating acid sites [46]. The quantification of the acid sites was carried out from the NH_3_-TPD experiments (Table 1). These data reveal how the number of acid sites can be considered as negligible with only 13 μmol g^−1^ for the Si sample. The incorporation of the Zr-species in the porous silica causes a progressive increase in the number of the acid sites when Zr-content of the porous silicas also increases, attaining a maximum number of acid sites of 524 μmol g^−1^ for the SiZr5 sample. According to previous research, the strength of these sites is variable although most of them are weak and moderate [46]. In this sense, previous research has reported that the presence of acid sites has a beneficial effect in the adsorption capacity of some biomolecules. However, these authors established that the number of acid sites must be modulated since an excess of acid sites can hinder the desorption of the biomolecule under mild conditions since the use of stronger conditions could cause the denaturalization of these biomolecules [46,65,66,67,68]. In addition, the incorporation of a high proportion of heteroatoms in the synthetic step causes an impoverishment of the textural properties in such a way the amount of available active sites and subsequent adsorption capacity also decrease [46].

The chemical composition of the adsorbents on their surface was analyzed by XPS (Table 2). The Si 2*p* core-level spectra only display a contribution located about 103.4 eV, which is ascribed to silicon species in the form of silica. In the case of the O 1*s* core-level spectra, it can be observed as a single contribution located at 532.9 eV attributed to Si–O–Si. The incorporation of the Zr-species causes a modification of the O 1*s* core-level spectra since the main contribution is shifted at a higher binding energy, about 533.2 eV. In addition, another contribution, with lower intensity, located at about 531.1 eV, appears. This contribution increases with the Zr-content, which is assigned to Si–O–Zr. The formation of segregate ZrO_2_ must be discarded since this contribution should appear at about 529.0 eV [92]. In the same way, the Zr 3*d*_5/2_ core-level spectra only show a contribution that appears at 182.8 eV [92,93], which is also higher than the bulk ZrO_2_ (182.2 eV) [93]. This fact confirms the incorporation of the Zr-species on the silica framework. The higher binding energy for the band located in the Zr 3*d*_5/2_ core-level spectra in the Zr-doped mesoporous silicas implies a higher ionic character of the bonding Zr–O and, therefore, involving a higher positive charge over Zr, which leads to the formation of Lewis acidity in these materials, as was indicated from NH_3_-TPD (Table 1) [94].

With regard to the molar ratio, it can be observed how (Si+Zr)/O molar ratio is close to 2.0 in all cases, which is in agreement with the theoretical value of SiO_2_ and ZrO_2_. However, the Si/Zr molar ratio is very far to the theoretical values since the concentration of the Zr on the surface is much lower than that expected due to the different hydrolysis rate of each alkoxide.

It is well-known that the pH is a key parameter in most adsorption processes. Considering this premise, the point of zero charge (pH_zpc_) was evaluated to discern the pH at which these porous materials reach a total zero charge. According to the data reported in Appendix A, it can be observed how Si sample is neutrally charged until pH = 4, indicating that Si–OH and Si–O–Si do not accept or give up H^+^-species below that pH, which agrees with the literature [46,95]. The analysis of the pH_zpc_ at high pH shows how the pH_f_–pH_i_ reaches positive values. This fact supposes that the release of H^+^-species from the adsorbents from pH > 4, obtaining a negatively charged surface by the formation of Si–O^−^ species [45]. The incorporation of the Zr-species in the synthetic step favors the formation of positive values (pH_f_–pH_i_) even at a pH lower than the porous silica.

From these data, it can be inferred that the Zr–OH species are more prone to lose H^+^-species than Si–OH. The study of the pH_zpc_ from pH = 9 show a clear decrease in the pH_f_–pH_i_ probably due to the partial leaching of the silica under pH conditions [46]. The pH_zpc_ studies seems to indicate that all porous silicas of this research have a predominance of negative charges in most pH ranges so it is expected that these porous materials can display higher affinity by the cationic groups of the biomolecules.

## 3. Batch Adsorption Tests

### 3.1. Effect of pH Buffers Solutions

It is well-known in the literature that several parameters such as pH, temperature or ionic strength have a strong influence on the protein adsorption [40,46,96,97,98]. The adsorption studies using SBA-15 as an absorbent have reported that the maximum adsorption capacity takes place at a pH value very close to the pI of the biomolecule due to a global balance, between the attractive protein–surface interaction and a repulsive protein–protein interaction, is usually the most optimal to obtain the highest adsorption values [98,99]. Under these pH conditions, the low electrostatic repulsions between the molecules in their pI facilitate the dense packaging of the adsorbate molecules. While the apparent total charge of the protein disappears in the pI, its surface still contains traces of positively and negatively charged amino acid residues, which interact by attraction with the negatively charged surface of the adsorbent [65,100]. However, at pH values away from the pI, the proteins will repel each other and thus cause a less compact packaging density at the adsorption surface [70].

Appendix A show the influence of the pH on the adsorption of IgG and HSA on the Si sample while Appendix A show how the IgG and HSA adsorption capacity on the SiZr5 sample. These adsorbents were selected due to these samples were those whose number of acid sites was lower and higher, respectively.

In the case of the IgG, whose pI is in the range 6.3–9.0, Appendix A display a strong dependence of the pH since the IgG adsorbed was about 200 mg g^−1^ for the Si sample, while the SiZr5 sample only attains an adsorption capacity of 100 mg g^−1^. These data indicate that the increase in acid sites worsen the adsorption capacity. This fact could be attributed to several parameters such as the incorporation of Zr-species, which causes a decrease in the pore volume and pore size as was indicated previously (Table 1), but also by the modification of the electronic density of the adsorbent, which could generate a greater number of repulsive interactions with the IgG. In previous research, it was reported that the highest adsorption capacity takes place when it is used pore expanded agents since the accessibility of biomolecules within the porous structure is much higher than those with a smaller pore diameter [45]. These authors also reported that the incorporation of a small proportion of Zr into the silica framework favors a higher amount adsorption of non-human biomolecules [46].

In both cases, it is noteworthy that those buffers work in a pH range similar to the pI (phosphate, MOPS, HEPES and TRIS-HCl) reaching similar adsorption values. However, acetate buffer, whose pH is below the pI, exhibits lower adsorption capacity although the adsorption of both biomolecules improved according as the pH of the acetate buffer also increased.

The HSA adsorption in both adsorbents, indicated previously (Si and SiZr5) (Appendix A), follows the same trend to that discussed in the IgG adsorption. Thus, the highest HSA adsorption was attained around its pI, reaching an HSA adsorption capacity around 183 and 163 mg g^−1^, respectively, for Si and SiZr5 samples using an acetate buffer (pH 4.8). As the pH moves away from the HSA pI, the adsorption capacity also diminishes, as indicates HSA adsorption results obtained using phosphate, MOPS, HEPES and TRIS-HCl buffers.

Focusing on an acetate buffer, the increase in the acidity of the porous silica does not have a clear influence in the amount of HSA adsorbed at pH 4.8 (pI of the HSA). This decrease is ascribed to the electrostatic repulsions of the same charge signal of the protein (negative) and the porous silica (negative). Nonetheless, this decrease is not equally pronounced for both adsorbents since the adsorbent with the highest acidity (SiZr5) has an HSA adsorbed amount, about 2.21 to 3.14 times lower than that obtained for the Si sample when analyzing the other buffers (phosphate, MOPS, HEPES and TRIS-HCl). This fact can be ascribed to the formation of Zr–O–Si bonds, which increase the ionic character of the adsorbent by the presence of Zr-partially charged, leading to Lewis acidity in these adsorbents, which seems to have an adverse effect on its adsorption capacity [94].

### 3.2. Effect Contact Time

Between the buffers tested, acetate and phosphate buffers were selected to carry out both kinetic and adsorption isotherms studies (Figure 3). These buffers were selected due to their pH range being very close to the pI.

The kinetic studies show a fast decrease in IgG concentration in the Si sample when using an acetate buffer (Figure 3A), attaining the equilibrium conditions after 120 min. The incorporation of the Zr-species into the porous silica worsens the adsorption values probably since the presence of the Zr-species diminishes the specific surface area and pore volume as well as increases the number of acid sites, which seems to be an adverse effect in the adsorption capacity probably due to the increase in the ionic character of the adsorbent. In addition, it is required longer adsorption time to reach the equilibrium conditions (about 180 min). In the case of the IgG adsorption using phosphate as a buffer (Figure 3C), the equilibrium conditions were also obtained faster for the Si sample in comparison to the Zr-doped porous silicas. The kinetic studies also revealed that SiZr20 sample has an adsorption capacity that is comparable to the Si sample, although it is required higher contact time, around 420 min, to reach similar values.

For HSA adsorption using an acetate buffer (Figure 3B), it was observed that Si sample shows the fastest and highest adsorption, reaching the equilibrium conditions after only 30 min. The incorporation of Zr-species into the porous silica causes a decrease in the adsorption capacity. In addition, the kinetic study reveals that the incorporation of Zr-species requires longer adsorption times to reach the equilibrium conditions. In the case of the HSA adsorption on phosphate buffer (Figure 3D), the adsorption is less pronounced due to the adsorption takes place at pH away from pI. From these data, it is inferred that the same trend to that observed when acetate was used as buffer since Si samples reaches the equilibrium conditions more quickly than in the case of adsorbents with Zr incorporated in the porous silica structure.

In summary, from the kinetic studies, it can be concluded that faster equilibrium adsorption takes place when the Si adsorbent is used. This fact implies that adsorption via electrostatic interaction is faster than interactions with greater ionic character, which appear in adsorbents with Zr-incorporated into its structure.

### 3.3. Adsorption Isotherms

Figure 4 compiles the profiles of the IgG and HSA adsorption isotherms in all adsorbents using acetate (Figure 4A,B) and phosphate (Figure 4C,D) buffers. For the IgG isotherms using an acetate buffer (Figure 4A), as was observed previously, it can be easily seen that biomolecules were adsorbed in greater quantity in the porous silica without Zr, attaining an IgG adsorption around 500 mg g^−1^ while the adsorbents with zirconium incorporated into the structure hardly reaches an adsorption capacity of 150 mg g^−1^. From these data, it can be inferred that porous silica displays high adsorption capacity even when pH is far from the isoelectric point while the Zr-doped pore silica seems to be more sensible to changes in the pH.

When the adsorption occurs with protein diluted in phosphate buffer at pH 7.0 (Figure 4C), it can be easily noted a considerable increase in the amount of IgG adsorbed in all porous materials in comparison to those results obtained in acetate buffer at pH 4.8. Thus, in the case of the Si sample, it can be observed an increase in the IgG adsorbed from 500 to 700 m^2^ g^−1^, while the Zr-doped absorbent also increases the adsorption capacity (from 150 to 250–350 mg g^−1^) although this increase is less pronounced in comparison to the Si sample. From these data, it can be confirmed that the maximum adsorption capacity takes place at a pH very close to the pI due to the repulsive interactions being minimized. In the case of the Zr-doped silicas, the improvement of the adsorption capacity is lower because, despite increasing the adsorption capacity when approaching the pI, the increase in the ionic character of the adsorbent by the incorporation of Zr prevents a rise adsorption capacity more pronounced.

The HSA adsorption using a buffer solution of acetate at pH 4.8 is compiled in Figure 4B. These data reveal that the maximum adsorption capacity is obtained for the Zr-doped porous silica with the lower Zr-content (SiZr20). This fact was also observed for other non-human biomolecules such BSA previously [46]. These data suggest that the presence of a small proportion of acid sites seems to be a beneficial effect in the adsorption capacity of HSA. In addition, it can be observed how the adsorption profile of this adsorbent shows high squareness. This fact suggests that HSA is strongly adsorbed on the Zr-doped porous silica with the lower Si/Zr molar ratio.

HSA adsorption in the porous silicas using a phosphate buffer at pH 7.0 (Figure 4D) was lower than 250 mg g^−1^ in all cases. Comparing this result with HSA adsorption using acetate, it can be noted a lower squareness of the adsorption isotherms so the interaction between HSA and the adsorbent must be lower than that observed when a pH of 4.8 was used. Considering that pH of the adsorption test is away to the HSA pI, the adsorption capacity, as well as the strength of the interaction adsorbent-HSA, are lower to than shown when acetate buffer. This fact is ascribed to the existence of repulsive interactions between the porous silica that is negatively charged and the HSA, which is also negatively charged. In addition, it is interesting to note that, when working on the HSA pI, SiZr20 was the sample with highest HSA adsorption capacity but, when the pH was far from pI, the SiZr20 sample provides the lowest adsorption values because of repulsive interactions. These data indicate that the incorporation of the Zr-species, which increases the ionic character of the adsorbent, seems to have an adverse effect in its adsorption capacity.

Appendix A present the fitting parameters obtained by Langmuir and Langmuir–Freundlich models for the HSA and IgG adsorption in all porous silicas studied at pH = 4.9 and 7.0, respectively.

For IgG adsorption using an acetate buffer at pH = 4.8 (Appendix A), in spite of the adsorption taking place at a pH away from the pI, the Si sample reaches a q_max_ much higher than other adsorbents; however, the Zr-based adsorbent is less prone to interact with IgG molecules under these pH conditions. Both the *k_L_* and *k_LF_* values provide interesting information about the strength of the interaction adsorbate–adsorbent. Thus, the highest *k_L_* and *k_LF_* values were obtained for the Si sample. This confirms a stronger interaction between the adsorbate and adsorbent, so the Zr-species seems to be an adverse effect in the strength of the interaction with the protein. From these data, it can be inferred that the more ionic character of the Zr-doped porous silica has an adverse effect in the IgG adsorption at a pH of 4.8. These data also suggest that the predominant adsorption must be via electrostatic interactions between polar groups (-NH_2_ or -COO) with hydrogen atoms and/or non-electrostatic interactions (van der Walls or London forces) [101].

As can be seen in Appendix A, the highest HSA adsorption capacity using an acetate buffer was obtained for the adsorbent with low zirconium content (SiZr20), although HSA adsorption capacity is very close to the Si sample since the q_max_ of both are very close between them in such a way the best adsorbent depends on the adsorption model. As the adsorption isotherm of the Si sample displays higher squareness, this adsorbent reaches the highest *k_L_* and *k_LF_* values. This fact supposes that the strongest interaction takes place with the Si sample. The addition of a low proportion of the Zr causes a decrease in the *k_L_* and *k_LF_* values in such a way the interaction of the HSA is also weaker, as was observed for the IgG using an acetate buffer.

For the IgG using a phosphate buffer at pH = 7.0 (Appendix A), the highest adsorption capacity was obtained for the adsorbent without zirconium (Si sample). In the same way, these data also reveal that this adsorbent also shows the strongest interaction with the IgG, being even higher than that estimated by the HSA adsorption due to the pI of IgG is closer to the pH range of the phosphate buffer. It is noteworthy that the change of the pH does not seem to affect IgG adsorption capacity on the Si sample greatly while the Zr-doped porous silica seems to be more sensitive to pH modifications because of the modification of the electronic density of the adsorbent by the incorporation of the Zr-species.

In the case of the HSA adsorption using a phosphate buffer (pH 7.0) (Appendix A), the highest adsorption capacity was obtained for SiZr20 and SiZr10 samples. Both the k_L_ and k_LF_ parameters reveal that the incorporation of a low amount of zirconium into the structure causes an increment of these parameters so it is expected a stronger interaction of these adsorbents with HSA. As we move away from the pI, a slight increase in acidity seems to have a positive effect on the adsorption capacity, although in any case it is much lower than that reached close to pI.

### 3.4. Characterization of the Samples after the Adsorption Test

In order to confirm the adsorption process, the sample after the IgG and HSA adsorption on the Si sample using a phosphate buffer were recovered and then these samples were characterized by a FT–IR, elemental analysis and XPS.

The FT–IR spectra of the Si-adsorbent after the IgG or HSA adsorption (Appendix A) shows the presence of new bands. The clearest changes are observed between 1600–1350 cm^−1^. Thus, it is noticeable the presence of two bands and the signals located at 1460 and 1405 cm^−1^, which are assigned to –C–H bending vibration modes [46]. In addition, it can also be observed the presence of another band located at about 1530 cm^−1^, which is attributed to the presence of –N–H and –C–H deformations [46]. The CHN analysis also showed the presence of both carbon and nitrogen species on the porous silica. To carry out these analyses, the samples with the highest q_eq_ were selected. These data reveal that, after the IgG adsorption, a C-content of 3.7% and an N-content of 0.9% were observed while, after the HSA adsorption, it observed a C-content of 1.2% and an N-content of 0.3%.

The analysis of the adsorbent after the adsorption process of HSA and IgG by XPS (Appendix A) shows a clear modification of the C 1*s* and N 1*s* core-level spectra. With regard to the C 1s core-level spectra, it can be observed a clear increase in their concentrations on the surface of the adsorbent (Appendix A and Table 3) due to the increase in the signal located at about 284.8 eV [93], which overlaps the adventitious carbon and the C–C bonds. In addition, it is also noticeable the arising of a new contribution located at about 288 eV [93], which is typical of the carbonate and/or carboxylate species. In the same way, the N 1*s* core-level spectra also display a new and broad contribution located at about 399.8 eV (Appendix A) [93], which is ascribed to the existence of amine and amide-species. Considering these characterization data, it seems clear that both HSA and IgG are adsorbed in the porous silica.

## 4. Protein Purification

### 4.1. Binary Solution Diluted in TRIS/HCl at pH 9.0

Considering the adsorption studies (Appendix A and Table 4), in general, it was observed that the incorporation of zirconium in the porous silica was disadvantage for IgG adsorption. Therefore, the Si sample was chosen for the next batch of experiments for IgG purification. In addition, it was observed that when using a protein solution diluted in acetate and phosphate, no clear selectivity was found with any one adsorbent. Thus, for the purification tests, it was chosen to dilute the protein solution in a TRIS–HCl buffer at pH 9.0 to IgG purify using binary systems IgG–HSA. This buffer was chosen due to the IgG adsorption in the Si sample being close to 180 mg g^−1^. This value is extremely higher in comparison to the HSA adsorbed (below 10 mg g^−1^) (Appendix A).

Under these conditions, both HSA and the adsorbent are negatively charged on their surface, so the repulsive interactions are predominant leading to lower adsorption values. However, the IgG protein has a pI between 6.3 and 9.0, so IgG presents a neutral charge in these conditions and the adsorption was more effective than HSA. This fact supposes that the IgG adsorption capacity was close to 700 mg g^−1^ while the HSA adsorption was only 30 mg g^−1^ (Appendix A). The fitting of the adsorption isotherms to the Langmuir and Langmuir–Freundlich models confirm the high selectivity towards IgG under these pH conditions as a consequence of the stronger interaction of the IgG with the adsorbent in comparison to the HSA, where the interaction adsorbate–adsorbent was very low and very linear.

For the next step, it was chosen to work with a binary system containing IgG and HSA with different concentrations. Figure 5 shows the adsorption profiles evaluated of three cases using binary mixtures and the single component profiles containing only IgG or HSA Si sample. By analyzing the performance of both adsorbents, it was possible to observe that an increase in the IgG proportion favors a rise in the adsorption capacity considerably. From these data, it can be deduced that IgG offers a greater interaction for Si adsorbent since the pH used in this study is close to the pI of the IgG in comparison to HSA.

From the adsorption data reported in Figure 5, there should be a competitive adsorption between IgG and HSA by the available active sites. However, the adsorption is preferentially towards IgG. Possible arguments for such low HSA adsorption may be due to the significant electrostatic repulsions between the same signal charge between the Si sample and HSA, since both presented negative surface charge at pH = 9.0. In addition, it has been reported that pH changes cause different conformations of HSA that can also have an adverse effect in the HSA adsorption of the porous silica [76].

If it is only considered the dimensions of the proteins, HSA (7.5 nm × 6.5 nm × 4.0 nm) [76] has lower dimensions in comparison to IgG (14.5 nm × 8.5 nm × 4.0 nm) [84]. From these data, it should be expected that HSA can more easily access the pores of the Si adsorbent. However, the higher adsorption capacity takes place towards IgG. The obtained data can be in agreement with the “Vroman effect” where low molecular weight proteins initially bind first to the surface of the adsorbent, but bulkier IgG protein molecules tend to expel part of these proteins with lower molecular weight leading to a more stable bond with the surface of the adsorbent [102,103,104]. In this sense, it must be remarked that the use of binary systems leads to adsorption isotherms with a higher linearity in comparison to the respective monocomponent isotherms. This fact could be ascribed to a lower interaction between adsorbate–adsorbent.

In previous work, Dahman et al. carried out adsorption studies of BSA (Bovine Albumin) and BHb (Bovine Hemoglobin) proteins in a binary solution using an adsorbent polystyrene based on an anion exchange resin [105]. These authors pointed out that the amounts adsorbed for each protein in a single component solution were very close between them. However, the analysis of these proteins in a binary solution, following the same cases reported in the present research (I, II and III), revealed that one protein affects the adsorption capacity of the other one. These authors suggested the presence of interactions between proteins in such a way that agglomerates of biomolecules are generated, which may not be able to access the porous structure due to steric effects [105]. Moreover, these authors mentioned that apparently the presence of both proteins caused a steric effect on adsorption probably due to competition for the surface of the adsorbent and, in addition, the interactions between the protein molecules in solution [105]. Considering these premises, it is expected that the interaction between proteins as well as the steric effect can have a key influence in the strength of the adsorbate–adsorbent.

In the same way, the adsorption data reveals that the presence of low proportion (case I and II) HSA seems to be an adverse effect in the IgG adsorption, a greater steric effect, as well as repulsive interactions between the adsorbent and the HSA since both are negatively charged. However, in case III, where the proportion of IgG is higher, the adsorption can be approached to that shown by the adsorption of its monocomponent, although the case III adsorption isotherm is much more linear, so the adsorbent–adsorbate strength must be lower than the monocomponent.

These arguments can be corroborated with the electrophoresis results of each test performed. Figure 6 shows the results obtained from the batch experiments using all cases of binary solution in the Si adsorbent. It can be observed that the first well was selected for the protein standard obtained from Sigma (HMW), the following wells were from the adsorption assay samples with the different concentrations at the beginning and at the end of the adsorption. The samples analyzed after adsorption represented the proteins in the liquid phase, and consequently, a decrease in the band of a certain protein denoted that was retained in the adsorbent.

From Figure 5, it can be observed clearly in the concentrations 1.0–4.0 mg mL^−1^ of the binary solution that IgG was more adsorbed, as indicated by the significant decrease in the band intensity of IgG corresponding to those concentrations. However, the intensity of the human albumin band (HSA) only displays a slight decrease. These findings can be observed in the three cases studied, where, as the proportion of IgG increases in the binary solution (Case I > Case II > Case III) the IgG band intensity always decreases, IgG presents a greater interaction with the Si adsorbent, so IgG is the preferentially adsorbed protein.

### 4.2. Human Serum Solution Diluted in TRIS/HCl at pH 9.0

Figure 7 shows the protein adsorption profiles present in human serum that were adsorbed on the Si sample. Human serum samples present: human albumin (HSA), around 60%, IgG, transferrin and other proteins with different molecular weights. Despite the human serum samples present many proteins besides HSA and IgG, the adsorption capacity is still pronounced since the IgG adsorption is about 300 mg g^−1^ using the Si adsorbent. Nonetheless, this value is about three times lower than that shown for the monocomponent adsorption test, which only contained IgG. This fact confirms that there is a greater competition between the innumerable human proteins to interact with the adsorbent surface. However, the IgG is much higher than the rest of the proteins due to the pH of the adsorption test (9.0) was close to the pI, so the repulsive effect between the Si-adsorbent and the IgG is much lower than that shown for other proteins whose pI is very far from the pH of the TRIS/HCl buffer. As was indicated previously, this adsorption isotherms are also much more linear than that shown for its respective monocomponent (Appendix A), confirming that the competition of several proteins by the adsorption sites has an adverse effect in the strength of the interaction adsorbate-adsorbent.

Similar to the binary HSA–IgG adsorption tests, the electrophoresis analysis can provide interesting information to confirm the protein adsorption using human serum solutions. According to electrophoresis from Figure 8, it was possible to observe which proteins had the highest concentration in the liquid phase and consequently, lower adsorption on the adsorbent. The band corresponding to the HSA molecular weight (66 KDa) was extremely concentrated in most of the samples analyzed. This fact is expected since HSA is the protein with the greatest quantity in human serum. However, other proteins with lower concentrations in human serum and molecular weights close to those of HSA may be masked in this region, so changes in the same region do not indicate that only HSA is being adsorbed but could also adsorb other proteins located in that same region. Analyzing Figure 8, it is possible to note that the region corresponding to the IgG band (molecular weight of 150 KDa) had a considerable decrease in its intensity showing that IgG was one of the main proteins adsorbed in the assays with the human serum in the Si-adsorbent.

## 5. Materials and Methods

### 5.1. Reagents

The reagents used in the synthesis of the porous silica were tetraethylorthosilicate (TEOS) (Sigma-Aldrich, St. Louis, MO, USA), zirconium(IV) propoxide in *n*-propanol ≥70% (*w*/*w*) (Alfa-Aesar), copolymer triblock Pluronic P123 (PEO20PEO_70_PEO20) (Sigma-Aldrich), 1,3,5-trimethylbenzene (TMB) (Sigma-Aldrich), NH_4_F (Vetec) and hydrochloric acid (37%) (VWR), while the reagents employed in the adsorption processes were human Immunoglobulin (IgG), human serum albumin (HSA), human serum from type AB supplied by Aldrich in all cases, morpholinopropane sulfonic acid (MOPS) (Aldrich), hydroxyethyl piperazine ethane sulfonic acid (HEPES) (Aldrich), Trizma-base (Tris) (Aldrich) and Coomassie Brilliant Blue (Aldrich). The following chemicals used in sodium dodecyl sulphate–polyacrylamide gel electrophoresis (SDS–PAGE) analysis were supplied by Sigma: acrylamide, bis-acrylamide, SDS and dithiothreitol. In all cases, the chemicals were of analytical grade and were used as received without any further purification. The electrophoresis calibration kit (High Molecular Weight—HMW) used for determining the molecular mass of various proteins (myosin, 212 kDa; α2-macroglobulin, 170 kDa; β-galactosidase, 116 kDA; transferrin, 76 kDa; glutamic dehydrogenase, 53 kDa) was provided by GE Healthcare. The water used to prepare buffers and other solutions was ultrapure (Milli-Q System, Millipore, MA, USA).

The gases employed were He (Air Liquide 99.999%, Paris, France), N_2_ (Air Liquide 99.9999%, Paris, France) and NH_3_ (Air Liquide 99.99%, Paris, France).

### 5.2. Synthesis of the Porous Silicas

The synthesis of the porous silicas was carried out following the methodology proposed by Zhang et al. [106] to obtain mesoporous silicas by a hydrothermal treatment although with some modifications. In a typical synthesis, P_123_ and NH_4_F were dissolved in 1.7 M HCl (aq.) at room temperature and then TMB, used as a swelling agent, was also added into the transparent solution to increase the dimensions of the micelle. Later, TEOS, used as a silicon source, was added dropwise and stirred for 24 h at the same temperature and then was transferred to a Teflon lined reactor and heated at 120 °C for 72 h. The final molar ratio in the synthesis gel was P123/SiO_2_/TMB/HCl/NH_4_F/H_2_O = 1/55/48/350/1.8/11100.

The incorporation of a small proportion of Zr into the mesoporous silica was performed following the same methodology described above [40,100] considering only that the Zr-species were added in the form of zirconium propoxide together with the Si-source in adequate proportions to reach Si/Zr molar ratios of 20, 10 and 5, respectively. The final molar ratio in the synthesis gel was P123/SiO_2_/ZrO_2_/TMB/HCl/NH_4_F/H_2_O = 1/52.38-45.83/2.62-9.17/48/350/1.8/11100. In the final molar ratio, it is necessary to in-dicate that the SiO2-ZrO2 molar ratio is variable to obtain Si/Zr molar ratio of 20, 10 and 5.

In all cases, the obtained gels were filtered, washed with water and dried at 60 °C for 12 h. Then, all samples were calcined at 550 °C for 6 h to remove the organic matter with a rate of 1 °C min^−1^ in an oven Selecta, SEL-HORN-R3L. The adsorbents were denoted as Si and SiZrX, being X the Si/Zr molar ratio.

### 5.3. Characterization of the Adsorbents

X-ray diffraction (XRD) analyses were carried out using an X’Pert Pro MPD diffractometer. This equipment displays a X’Celerator detector and a Ge (1 1 1) primary monochromator (strictly monochromatic Cu-Kα radiation). Long range ordering of the synthesized materials was carried out by small angle X-ray diffraction between 2θ (1–10°).

The analysis by transmission electronic microscopy (TEM) was carried to study the morphology of the porous materials. These analyses were performed using a Philips CM 100 Supertwin-DX4 microscope. Prior to the study, the porous materials were dispersed in ethanol, and they were put on a Cu grid (300 mesh).

Fourier-transform Infrared (FT–IR) spectra were carried using FTIR 8400S supplied by Shimadzu. This equipment has a standard mid-IR DTGS detector. The background spectrum was performed with KBr. The spectra were collected in a range of wavenumber of 400–4000 cm^−1^.

The textural properties of the porous materials were determined by N_2_ adsorption-desorption isotherms at −196 °C using an automatic ASAP 2000 supplied by Micromeritics. Prior the N_2_ adsorption–desorption measurements, the samples were outgassed at 150 °C and 10^−4^ mbar for 12 h. The specific surface area was determined from the Brunauer–Emmett–Teller (BET) equation considering a N_2_ molecule cross section of 16.2 Å^2^ [89]. The pore size distribution (D_p_) and total pore volume (V_p_) were estimated considering the desorption branch of the isotherm through the Non-local Density Functional Theory (NLDFT) [91].

The total acidity of the adsorbents was estimated from ammonia thermo-programmed desorption (NH_3_-TPD) profiles. The materials were previously cleaned under He flow from room temperature to 550 °C (10 min) in an oven supplied by Hobersal, and subsequently cooled to 100 °C. Gas He flow (35 mL min^−1^) was then passed to eliminate the physisorbed ammonia. Finally, thermo programmed desorption was carried out by heating the samples from 100 to 550 °C at a heating rate of 10 °C min^−1^ in an oven supplied by Hobersal. The evolved ammonia in the desorption process was analyzed by an on-line TCD.

X-ray photoelectron spectra were collected using a Physical Electronics PHI 5700 spectrometer with non-monochromatic Mg Kα radiation (300 W, 15 kV, and 1253.6 eV) with a multi-channel detector. Spectra of pelletized samples were recorded in the constant pass energy mode at 29.35 eV, using a 720 μm diameter analysis area. Charge referencing was measured against adventitious carbon (C 1*s* at 284.8 eV). A PHI ACCESS ESCA-V6.0 F software package was used for acquisition and data analysis. A Shirley-type background was subtracted from the signals. Recorded spectra were always fitted using Gaussian–Lorentzian curves in order to determine the binding energies of the different element core levels more accurately.

The zero-point charge (pH_ZPC_) of the adsorbent was evaluated in a range of pH (4–12), using 50 mL of NaCl solutions (0.01 N) in contact with 15 mg of materials [107]. The pH of the NaCl solutions were previously set with the addition of acid (HCl) and basic (NaOH) solutions. After 48 h, the difference between equilibrium and initial pH was recorded.

### 5.4. Adsorption Batch Experiments

The influence of the pH in the adsorption capacity was performed considering several buffer solutions (Acetate, Phosphate, MOPS, HEPES and TRIS-HCl) were used. In addition, adsorption isotherms IgG or HSA were measured using mesoporous material with different Si/Zr molar ratio as adsorbent. Batch experiments were performed in duplicate at 25 °C. In each experiment, 15 mg of sample was mixed with 3.0 mL of the protein solution using acrylic tubes. These tubes were stirred in an orbital shaker (Tecnal TE-165, Brazil) for the required time to attain the equilibrium conditions. It was prepared as a buffer solution containing IgG or HSA by dissolving these proteins (1.0 mg mL^−1^) in 25 mM of buffer solutions with pH ranges: 4.0–5.6 for acetate buffer; 6.0–8.0 for phosphate buffer; 6.5–7.9 for MOPS buffer; 6.8–8.0 for HEPES buffer; 7.2–9.0 for TRIS–HCl buffer.

An initial concentration of 1.0 mg mL^−1^ was prepared for the kinetic adsorption studies for all adsorbents in order to determine the equilibrium time. The adsorption isotherms were carried out considering initial concentrations of IgG and HSA between 1.0 and 6.0 mg mL^−1^. In all cases, the supernatant was collected and centrifuged for 10 min at 10,000 rpm (refrigerated microcentrifuge Cientec CT-15000R). The protein concentration in the liquid phase (supernatant) was determined by UV–Vis light absorbance at wavelength of 280 nm using a UV–Vis spectrophotometer Biomate 3, Thermo Scientific.

### 5.5. Batch Protein Purification with Binary Solution (IgG and HSA)

#### 5.5.1. Binary Solution Preparation

The binary solution containing IgG and HSA was prepared with three different ratios of mixture of IgG/HSA as follows:

**Case I.** 
*Proteins solutions containing IgG 25% and HSA 75%.*


**Case II.** 
*Proteins solutions containing IgG 50% and HSA 50%.*


**Case III.** 
*Proteins solutions containing IgG 75% and HSA 25%.*


For each case, it was prepared in six different binary protein concentrations (1.0, 2.0, 4.0, 5.0 and 6.0 mg mL^−1^) diluted in TRIS/HCl 25 mM buffer at pH 9.0. Appendix A shows all the concentrations that were prepared for each case studied, including the individual protein solution IgG or HSA. Firstly, it was prepared a solution containing IgG or HSA with different concentrations, and then was mixed in the same volume with these proteins, according to Appendix A.

The mass of protein adsorbed per mass of adsorbent (mg g^−1^) at equilibrium was calculated using the mass balance described by Equation (1):(1)$$q=\frac{V(C_{0}-C_{eq})}{m_{ads}}$$
where *V* (mL) is the volume of the sample solution, *C*_0_ and *C_eq_* (mg mL^−1^) are the protein concentrations before and after the adsorption, respectively, *q* (mg g^−1^) is the amount of protein adsorbed onto the adsorbent and *m_ads_* (g) is the mass of each adsorbent.

The adsorption isotherms were adjusted to the Langmuir (2) and Langmuir–Freundlich (3) models by the fitting of the experimental data by using commercial software (Origin^®^ software, Microcal, Northampton, MA, USA).
(2)$$q=\frac{q_{\max}k_{L}C_{eq}}{1+k_{L}C_{eq}}$$
(3)$$q=\frac{q_{\max}{\left(k_{LF}C_{eq}\right)}^{n}}{1+{\left(k_{LF}C_{eq}\right)}^{n}}$$
where *q_max_* (mg g^−1^) is the maximum adsorption capacity, *C_eq_* (mg mL^−1^) is the concentration of protein in solution at equilibrium, *K_L_* and *K_LF_* are the Langmuir and Langmuir–Freundlich constants, respectively, which are related to the affinity between protein and adsorbent, *n* is a coefficient for the Langmuir–Freundlich model, related to the heterogeneity of the adsorbent surface.

#### 5.5.2. Protein Quantification in Binary Solution

The protein concentration in the liquid phase (supernatant) was determined by the UV–Vis light absorbance at a wavelength of 280 nm (UV–Vis spectrophotometer Biomate 3, Thermo Scientific, Boston, MA, USA). For all cases I, II and III studied were realized calibration curves to obtain the concentration from binary solutions.

### 5.6. Batch Protein Purification with Human Serum Solution

#### 5.6.1. Human Serum Samples Diluted in TRIS/HCl pH 9.0

Human serum samples from Sigma showed concentration of about 50.0 mg mL^−1^ and, therefore, it was necessary to dilute in Tris/HCl buffer at pH 9.0 to obtain different concentration between 1.0 and 6.0 mg mL^−1^.

#### 5.6.2. Protein Quantification by Bradford Methods

The total protein content was analyzed by the Coomassie blue method according to the procedure described by Bradford [108] using the Bovine Serum Albumin (BSA) as the reference protein. The intensity of the color was measured at 595 nm using a UV–Visible spectrophotometer.

### 5.7. Electrophoresis Analyses

SDS–PAGE of protein samples was performed in 7.5% polyacrylamide gels under denaturing reducing conditions [109] using a Mini-Protean III System (Bio-Rad, Hercules, CA, USA). The runs were carried out at 150 V in 7.5% separation gels with a 4% stacking gel. Protein bands were developed by Coomassie Blue.

## 6. Conclusions

In the present research, it has been synthesized a set of porous silica, which have been modified to increase the pore diameter and reduce the length of the pore in order to favor the hosting of bulky biomolecules such as IgG and HSA. The incorporation of a heteroatom into the porous silica framework was another parameter evaluated. The analysis of the textural properties has revealed that the incorporation of Zr species negatively affects its textural properties, although these Zr sites also produce an increase in the acid sites.

The analysis of the adsorption capacity at different buffers has indicated that the maximum adsorption capacity takes place at a pH close to the pI, since under these pH conditions, the repulsive effects are minimized. The adsorption isotherm showed that the incorporation of the Zr species has an adverse effect on the adsorption capacity of HSA and IgG, mainly in the case of a SiZr10 and SiZr5 catalyst, probably due to these adsorbents have poorer textural properties than those shown for the Si catalyst.

Considering that the isolating and purification of the proteins is a key parameter so that these proteins can be sustainable, batch purification experiments were carried out. The study of the adsorption capacity in the Si sample using binary human serum solutions by electrophoresis SDS–PAGE revealed that this adsorbent has a considerable amount of adsorption for IgG, while the HSA adsorption can be considered as negligible. Therefore, in cases where the only objective is to concentrate IgG from a solution containing a mixture of other undesirable proteins, this adsorbent can be used successfully.

## Figures and Tables

**Figure 1 ijms-22-09164-f001:**
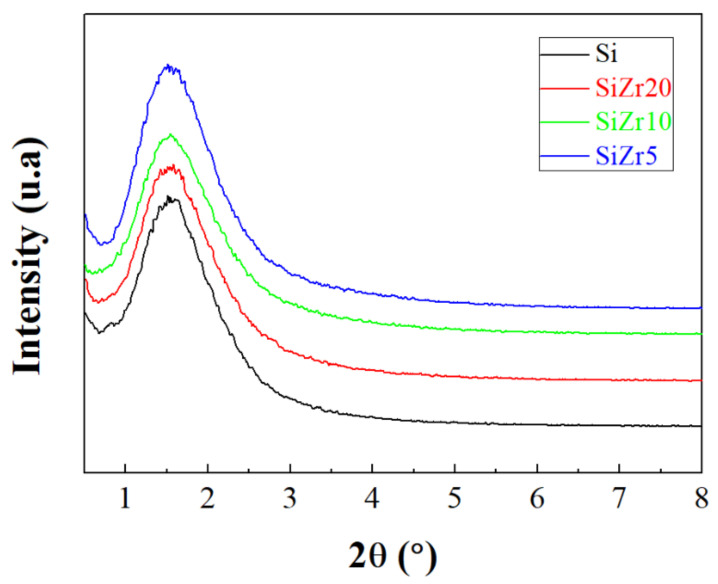
X-ray diffractograms of Si, SiZr20, SiZr10 and SiZr5 samples.

**Figure 2 ijms-22-09164-f002:**
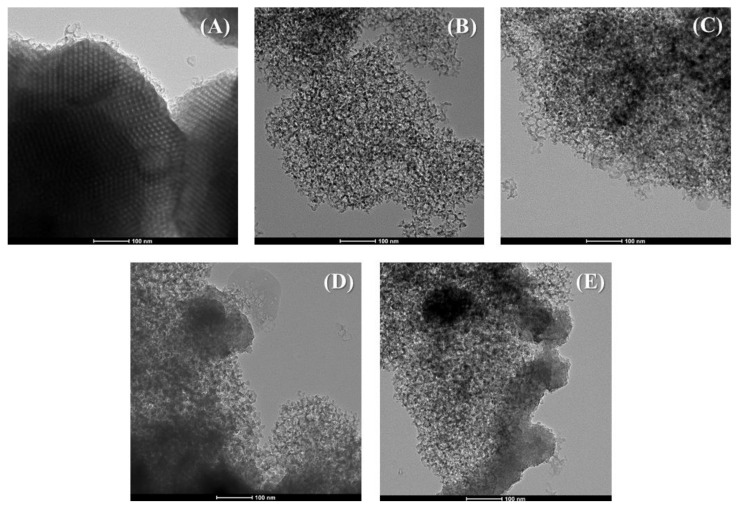
TEM micrographs of the traditional SBA-15 (**A**), Si (**B**), SiZr20 (**C**), SiZr10 (**D**) and SiZr5 (**E**) samples.

**Figure 3 ijms-22-09164-f003:**
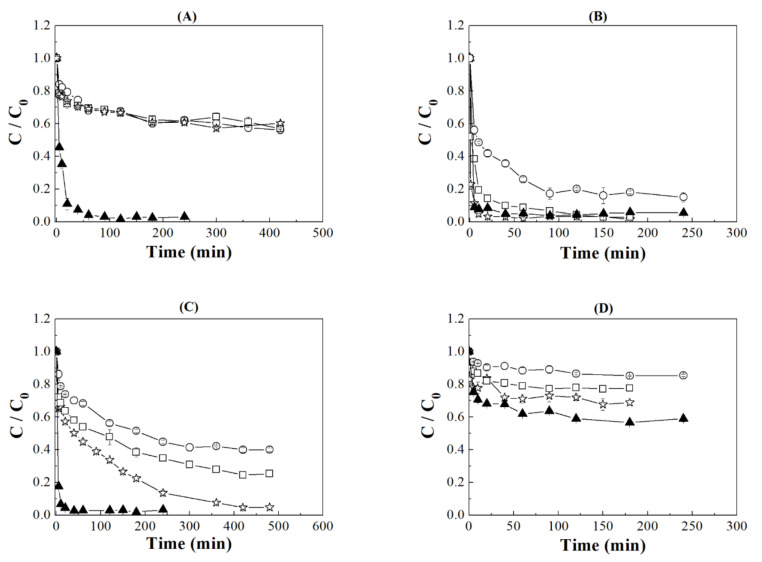
Kinetic profiles of IgG adsorption (**A**,**C**) and HSA (B,D) in: Si (▲), SiZr20 (*), SiZr10 (□) and SiZr5 (○) using acetate (**A**,**B**) buffer at pH 4,8 and phosphate (**C**,**D**) at pH 7.0. 

: the error of each experiment.

**Figure 4 ijms-22-09164-f004:**
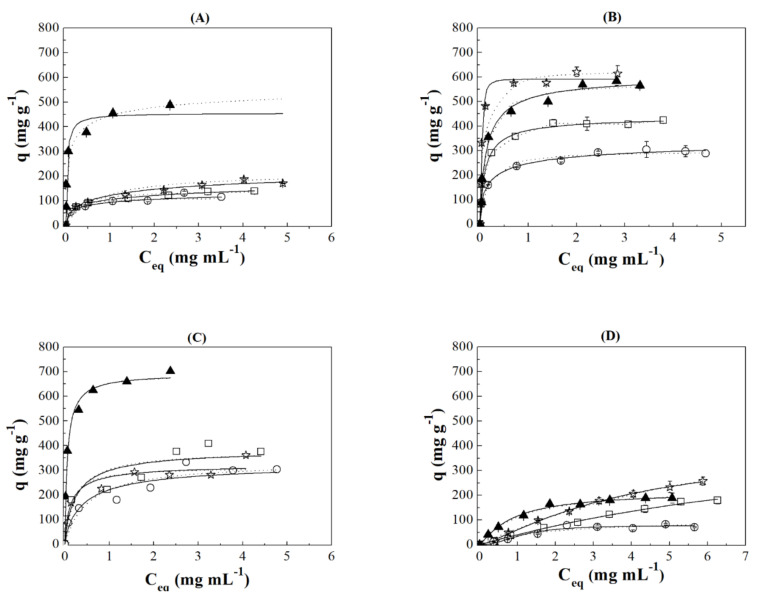
Adsorption isotherm of IgG (**A**,**C**) and HSA (B,D) in Si (▲), SiZr20 (*), SiZr10 (□) and SiZr5 (○) using an acetate buffer (**A**,**B**) at pH 4,8 and phosphate (**C**,**D**) at pH 7.0. 

: the error of each experiment.

**Figure 5 ijms-22-09164-f005:**
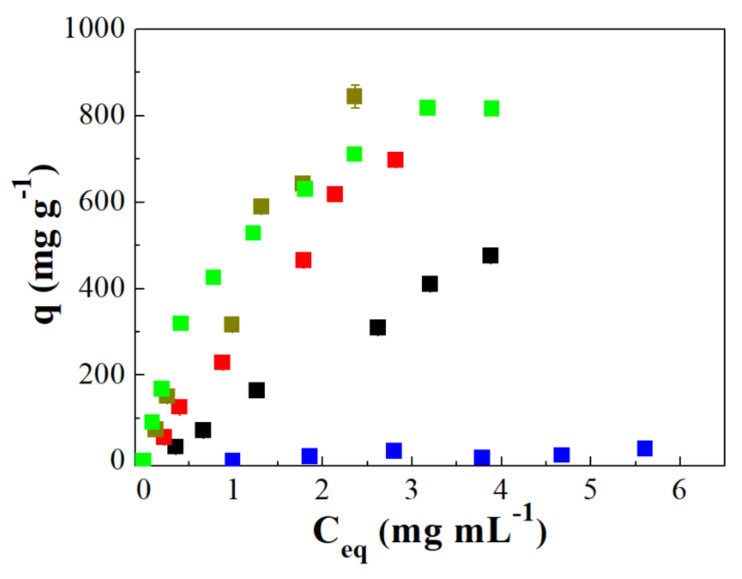
Batch adsorption experiments with individual protein solution containing IgG or HSA and solutions containing binary mixtures at three different cases. The conditions for all experiments were: 3.0 h of contact time, 3.0 mL of protein solution in contact with 15 mg of Si adsorbent. Case I (

), case II (

), case III (

), 100% of IgG (

) and 100% of HSA (

).

**Figure 6 ijms-22-09164-f006:**
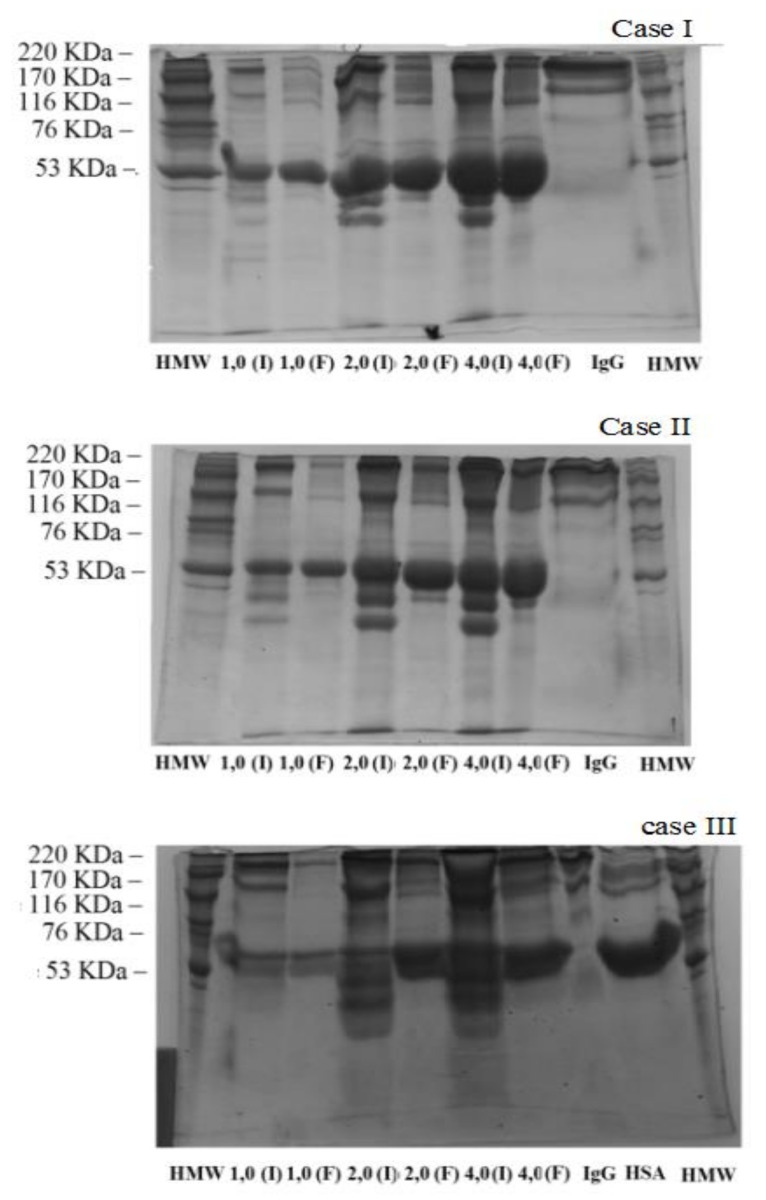
SDS–PAGE analysis of samples before and after adsorption batch experiments with binary solution diluted in Tris/HCl at pH 9.0 revealing with Coomassie brilliant blue for all. HMW (molecular weight marker: Myosin (220 KDa), α-macroglobulin (170 KDa), β-Galactosidase (116 KDa), transferrin (76 KDa) e Lutamic Dehidrogenase (53 KDa)), I (initial concentrations: 1.0, 2.0, 4.0 mg mL^−1^), F (final concentrations: 1.0, 2.0, 4.0 mg mL^−1^), IgG (IgG standard), HSA (HSA standard).

**Figure 7 ijms-22-09164-f007:**
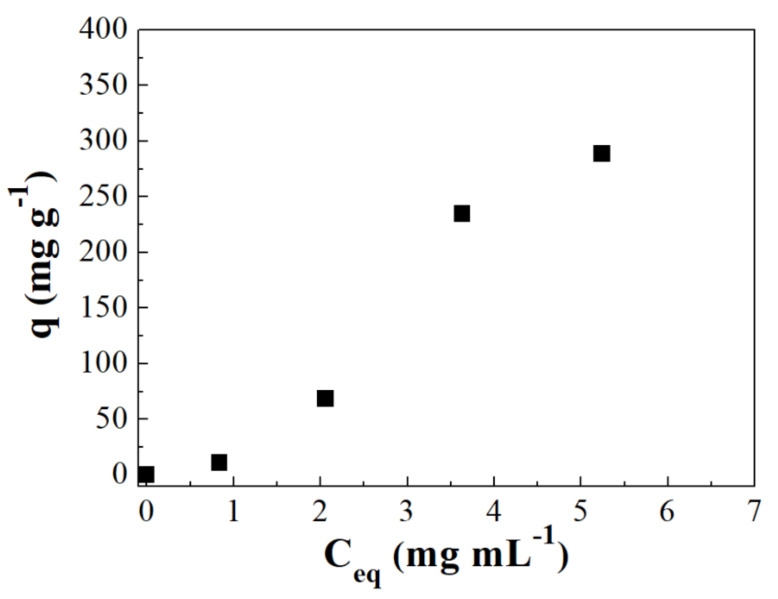
Batch experiments for human serum adsorption in SBA-15-FTMB. Contact time (4.0 h), mass of adsorbent (15 mg), volume of sample (3.0 mL) and buffer TRIS/HCl 25 mM at pH 9.0. Protein quantification by Bradford method.

**Figure 8 ijms-22-09164-f008:**
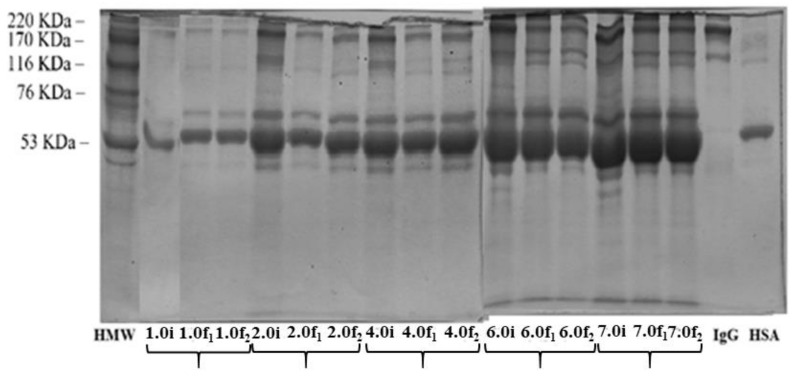
SDS–PAGE analysis of samples before and after adsorption batch experiments with human serum diluted in Tris/HCl at pH 9.0 revealing with Coomassie brilliant blue for all. HMW (molecular weight marker: Myosin (220 KDa), α-macroglobulin (170 KDa), β-Galactosidase (116 KDa), transferrin (76 KDa) e Lutamic Dehidrogenase (53 KDa)), X.0i (initial concentrations: 1.0, 2.0, 4.0, 6.0 and 7.0 mg mL^−1^), X.0f (final concentrations: 1.0, 2.0, 4.0, 6.0 and 7.0 mg mL^−1^), IgG (IgG standard), HSA (HSA standard). Each test was carried out in duplicated X.0f_1_ and X.0f_2_.

**Table 1 ijms-22-09164-t001:** Textural properties of the porous silicas and total acid sites estimated from NH_3_-TPD.

Sample	S_BET_ (m^2^ g^−1^) ^1^	V_P_ (cm^3^ g^−1^) ^2^	D_P_ (Å) ^3^	Total Acid Sites (μmol g^−1^) ^4^
Si	337	2.54	30.1	13
SiZr20	416	2.38	25.8	127
SiZr10	432	1.80	17.4	342
SiZr5	420	1.51	11.9	524

^1^ S_BET_: Specific surface area. ^2^ V_P_: Total pore volume. ^3^ D_P_: Pore diameter. ^4^ Total acid sites estimated from NH_3_-TPD experiments.

**Table 2 ijms-22-09164-t002:** Spectral parameters, estimated by XPS, of the porous silicas.

Sample	Atomic Concentration (%) *Binding Energy (eV)*				Molar Ratio	
	C 1s	O 1s	Si 2p	Zr 3d	(Si+Zr)/O	Si/Zr
Si	4.87% *284.8*	64.13% *532.8*	31.00% *103.4*	-	2.07	∞
SiZr20	5.45% *284.8*	60.93%/2.47% *533.2/531.2*	30.16% *103.3*	0.46% *182.7*	2.07	65.56
SiZr10	5.06% *284.8*	59.71%/3.01% *533.2/531.2*	31.63% *103.4*	0.56% *182.8*	1.95	56.48
SiZr5	4.94% *284.8*	57.82%/5.13% *533.2/531.2*	30.29% *103.3*	0.95% *182.8*	2.04	31.88

**Table 3 ijms-22-09164-t003:** Spectral parameters, estimated by XPS, of the porous Si sample before and after the adsorption process using HSA and IgG and a phosphate buffer at pH 7.0.

Sample	Atomic Concentration (%) *Binding Energy (eV)*			
	C 1s	O 1s	Si 2p	N 1s
**Si**	4.87% *284.8*	64.13% *532.8*	31.00% *103.4*	-
**Si (HSA)**	15.21%/4.49% *284.8/287.8*	50.18% *532.5*	24.94% *103.3*	5.18% *399.8*
**Si (IgG)**	16.47%/4.59% *284.8/287.7*	49.20% *532.6*	24.75% *103.4*	4.98% *399.7*

**Table 4 ijms-22-09164-t004:** Adjusted parameters by Langmuir and Langmuir–Freundlich (LF) for protein adsorption (HSA and IgG diluted in TRIS/HCl 25 mM at pH 9.0) in the Si-adsorbent.

Sample	HSA		IgG	
	L	LF	L	LF
*q_max_* (mg g^−1^)	50.42 ± 2.1	-	1.153.9 ± 70.2	936.02 ± 26.2
*k_L_* (mL mg^−1^)	0.032 ± 0.015	-	1.099 ± 0.15	-
*k_LF_* (mL mg^−1^)	-	-	-	1.53 ± 0.15
*R^2^*	0.52	-	0.99	0.99
*q_max_* (mg g^−1^)	-	-	-	1.23 ± 0.06

## Data Availability

All data associated with this study are provided in the main figures and tables and in the Appendix A.

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
