# Peer review of "Protein Adsorption onto Modified Porous Silica by Single and Binary Human Serum Protein Solutions"

_ijms, 2021, doi:10.3390/ijms22179164_

Round 1
Reviewer 1 Report
This is very interesting research article.
I would like to bring an attention to the authors to revise their manuscript based on my comments:
- I suggest to seeking professional editing of the text. There are flaws in grammar and style.
2) The ABSTRACT is not well structured. The abstract should adhere to international standards: introduction (one sentence), materials/methods, results, conclusions.
3) Statement in the INTRODUCTION section is insufficient. More detail explanations about the basic concept of new technology and new materials are necessary to claim the basic idea of this research for readers' understanding, regarding recent previous works about other DDS for nanoparticles application in biomedical engineering and drug delivery with quotation of their references is missing. It would be great if Authors revise their manuscript with such quotation of references list in the revised manuscript. Authors can use the following:
- Pharmaceutics, 12, 768-779 (2020)
-Polymers For Advanced Technologies, 30, 2647-2655 (2019)
- Polymers For Advanced Technologies, 26, 198-211 (2015)
- Acta Biomaterialia, 8, 4224-4232 (2012)
-Polymers For Advanced Technologies, 29, 2564-2573 (2018)
-Journal of Biomedical Materials Research Part A, 105, 2851-2864 (2017)
- International Journal of Pharmaceutics, 500, 162-178 (2016)
- Journal of Controlled Release, 206, 161-176 (2015)
- Polysaccharides: Bioactivity and Biotechnology, 1969-1990 (2015)
- Journal of Nanoscience and Nanotechnology, 14, 3328-3336 (2014)
- Polymers For Advanced Technologies, 25, 1216-1225 (2014)
- Journal of Nanoparticle Research, 15, 1345-1355 (2013)
- International Journal of Nanomedicine, 7, 4159-4168 (2012)
4) Materials and Methods:
-is there any further data that clearly shows the charge surface and of these porous silica
-what is the stability of these vectors after transfection of the cells?
Reviewer 2 Report
Dear Authors, dear Editors,
In the present manuscript, the Authors have produced porous silicates whose surface area has been modified to increase pore diameters while decreasing pore depth, all to be suitable as a carrier for containing bulky biomolecules such as IgG or HSA. The examinations in the manuscript are adequate. The figures and tables are informative, however, some additions need to be improved before publication, and a few modifications need to be considered.
- The introduction was very long (two pages), shortening is recommended. Since the authors in many cases use abbreviations, it would be advisable to include a list of abbreviations at the beginning of the manuscript.
- The first table shown in the manuscript begins on page 6 (named Table 2) and the analysis of the chemical composition on the surface of the absorbents by XPS. Table 1, which according to line 199 of the manuscript contains the values ​​of the special surface area (SBET values) of the materials later, is shown only on page 8. It is necessary to swap the position of the two tables.
-In the case of Figure 5, the authors provided their measurement results with colored but uniformly shaped markers. However, black squares appear in the caption, which does not allow the results to be identified.
-For Figure 6, where does the 0.5 mg/ml label appear in the SDS-Page analysis? In the caption, for the final concentration, mg is missing, only mL-1 is given as a unit of concentration.
-For Figure 8, where exactly is the result of the 6 and 7 mg/mL sample shown in the figure?
-In line 658 is missing the type of oven suitable for forming 550 C, the name of its manufacturer.
-There are 106 references for writing the manuscript. At the same time, the manuscript uses only 98 publications, and in the case of the last 8 references, no author, title or journal is given.
Author Response
Dear Authors, dear Editors,
In the present manuscript, the Authors have produced porous silicates whose surface area has been modified to increase pore diameters while decreasing pore depth, all to be suitable as a carrier for containing bulky biomolecules such as IgG or HSA. The examinations in the manuscript are adequate. The figures and tables are informative, however, some additions need to be improved before publication, and a few modifications need to be considered.
- The introduction was very long (two pages), shortening is recommended. Since the authors in many cases use abbreviations, it would be advisable to include a list of abbreviations at the beginning of the manuscript.
Following the advice of the reviewer, the introduction has been summarized although another reviewer also indicates that the authors must include some sentences and references in the manuscript.
- The first table shown in the manuscript begins on page 6 (named Table 2) and the analysis of the chemical composition on the surface of the absorbents by XPS. Table 1, which according to line 199 of the manuscript contains the values of the special surface area (SBET values) of the materials later, is shown only on page 8. It is necessary to swap the position of the two tables.
The authors thank the suggestion of the reviewer, the first table (Table 1) appears in page 5. In this tables appears the textural properties data and the acidity. Later, this Table 1 is also mentioned in Table 1. Then, spectral parameters obtained from XPS appears in page 6.
-In the case of Figure 5, the authors provided their measurement results with colored but uniformly shaped markers. However, black squares appear in the caption, which does not allow the results to be identified.
The authors have revised the captions and the squares appears with colors. Maybe, it is a problem of the word format. The next version will submitted in pdf version to avoid these problems.
-For Figure 6, where does the 0.5 mg/ml label appear in the SDS-Page analysis? In the caption, for the final concentration, mg is missing, only mL-1 is given as a unit of concentration.
According to the suggestion of the reviewer, the manuscript has been revised 0.5 mg/ml has been removed in the SDS-Page analysis. The first point is 1.0. In addition, the term mL-1 was corrected by mg mL-1.
-For Figure 8, where exactly is the result of the 6 and 7 mg/mL sample shown in the figure?
Following the advice of the reviewer, Figure 8 and its caption has been modified to observe the data of 6 mg/mL and 7 mg/mL.
-In line 658 is missing the type of oven suitable for forming 550 C, the name of its manufacturer.
The authors have indicated that the calcination of the organic matter was carried out in an oven supplied by Selectad while the acidity studies were performed in a Hobersal oven.
-There are 106 references for writing the manuscript. At the same time, the manuscript uses only 98 publications, and in the case of the last 8 references, no author, title or journal is given.
The authors thank the suggestion of the reviewer. The references have been corrected in the present revision.

Round 2
Reviewer 2 Report
Dear Authors, dear Editors,
the answers to my questions were given correctly by the authors. Manuscript errors were also corrected. For my part, there are no further questions.